# Intermittent colonisation with Methicillin-Resistant *Staphylococcal aureus* can be eradicated from the Airways of Adults with Cystic Fibrosis

**DOI:** 10.3390/antibiotics8030113

**Published:** 2019-08-09

**Authors:** Lucy Ranzenbacher, Yang Song, Alison Merchant, Peter G Middleton

**Affiliations:** 1Department of Respiratory and Sleep Medicine, Westmead Hospital, Westmead, NSW 2145, Australia; lucy.ranzenbacher@health.nsw.gov.au (L.R.); Yang.Song@health.nsw.gov.au (Y.S.); 2CF Research Group, Ludwig Engel Centre for Respiratory Research, Westmead Institute of Medical Research, Westmead, NSW 2145, Australia; Alison.Merchant@health.nsw.gov.au

**Keywords:** cystic fibrosis, MRSA, eradication

## Abstract

The airways of people with cystic fibrosis (CF) are chronically colonised with different pathogens. With recent interest in methicillin-resistant *Staphylococcus*
*aureus* (MRSA), we have recently examined the rates of MRSA colonisation in different groups within our CF Service. This paper now examines the effectiveness of eradication strategies to clear the MRSA colonisation.

## 1. Introduction

Cystic fibrosis (CF) is the most common lethal inherited disease in Australia. People with CF have characteristic colonisation of the airways with methicillin-sensitive *Staphylococcus aureus* (MSSA), methicillin-resistant *Staphylococcus aureus* (MRSA) and/or *Pseudomonas aeruginosa* (PsA). In our recent paper examining the risks of people with CF who were health care workers and in particular their risk of colonisation with MRSA, we collected data concerning eradication therapy, demographics, treatment details and outcomes [1]. Successful clearance rates of MRSA of up to 80% in early colonisation have been reported [2,3,4]. Our eradication process included both oral and intravenous antibiotics directed at the MRSA, together with topical mupirocin to the anterior nares to reduce local colonisation in the nasal passage.

## 2. Results

Over the period 2001–2015, a total of 252 patients were cared for at the Westmead Adult CF Service, Westmead NSW Australia. During this time, all sputum samples were cultured for the presence of MSSA, MRSA, *Pseudomonas aeruginosa* and other standard pathogens. Those who grew an organism on at least two occasions within 12 months were termed “colonised”. This was further divided into those who were termed intermittent, where <50% of the cultures were positive for the organism of concern (e.g., MRSA/MSSA/PsA), versus the chronic colonisations, where more than 80% of samples isolated the organism. The chronic and intermittent colonisation data is presented in Table 1.

Over the 2001–2015 period, a total of 20 adults with CF isolated MRSA on more than two occasions, of whom 12 were chronically colonised and eight were intermittently colonised. Further details of these MRSA colonised patients are shown in Table 2.

The summary of the characteristics as shown in Table 2 suggest that the intermittently colonised patients with CF were more likely to be successful in eradication. Out of the eight intermittently colonised patients, three were not offered eradication. Of the five patients who were offered MRSA eradication therapy, all five cleared MRSA. *P. aeruginosa* was the most common co-coloniser. Two out of the 20 patients were chronically colonised with other organisms, one with a coliform (not eradicated) and one with *Haemophilus influenzae* (eradicated). Both of these patients also grew PsA. There was a trend to better eradication of MRSA in those with concurrent PsA infection, but there was no statistical predictors for eradication: no difference in males versus females, older versus younger subjects, or co-colonisation with PsA. There was also no predilection for occupation or employment status, which has previously been discussed [1].

Out of the 20 patients colonised with MRSA, 16 were offered eradiation therapy. The four patients excluded from eradication therapy had previously shown non-compliance with other therapies and were deemed unsuitable for such a therapy regimen.

Of the 16 patients offered eradication therapy, eradication was successful in five of the five intermittently colonised subjects who were offered eradication therapy. Only three out of the eleven chronically colonised subjects achieved eradication.

The 16 medical records were examined with respect to dose and duration of oral and intravenous MRSA-targeted antibiotic therapy. Three records were incomplete and not suitable for analysis. The treatment records for these 13 remaining patients were analysed. Vancomycin was used intravenously whilst rifampicin and fucidin were oral preparations and the mucopiricin was a topical ointment. These details are displayed in Table 3.

## 3. Discussion

MRSA colonisation in the adult CF population is associated with poor outcomes [5,6]. The clearance of MRSA colonisation/infection from the airways of adults with CF is likely to be of assistance in the maintenance of good health. Eradication protocols vary between different clinics around the world. In general, the protocol used here included long-term rifampicin and fucidin, as described previously [7]. Difficulties with eradication protocols include drug interactions, side effects and toxicities [8]. The current experience with adherence also reflects the difficulties with long term complicated treatment protocols.

Successful eradication was achieved in those with intermittent colonisation and this likely reflects early infection with MRSA. Successful early clearance has been described previously [9], with clearance rates of ~80% [2,3,4]. Of the individuals where eradication was successful, this typically occurred earlier in the course of treatment, and was reflected in a shorter duration of rifampicin/fucidin/mucopiricin.

Intravenous vancomycin was used in the majority of adults with MRSA colonisation who were symptomatic. Whilst the use of this agent did not appear to improve eradication, it was effective at clinical outcomes, suppressing the reaction to the MRSA colonisation.

PsA and MRSA co-infection occur in 60% of our population. Co-infection with these organisms is associated with a more rapid deterioration of lung function [10]. The challenges achieving eradication of chronic PsA colonisation in people with CF are widely described [11,12,13]. Given the poorer outcomes for co-infection and challenges with PsA eradication, we would therefore encourage eradication therapy targeting MRSA.

One individual underwent 2 prolonged periods of therapy directed at eradicating MRSA, which were unsuccessful. However, following extensive surgical management of his nasal polyposis, a further course of eradication therapy was successful. Recently, a combination of oral therapy and topical mupirocin ointment was described in those patients who were newly colonised with MRSA [4] demonstrating the diversity of eradication protocols in different clinics.

## 4. Materials and Methods

The Australian CF Data Registry (ACFDR) for the Westmead Clinic was searched to identify all adults who attended the clinic between 2001 and 2015. Demographic data on the patients including age, gender, FEV_1_, sputum colonisation and occupation were collected. Manual assessment of those with MRSA colonisation and the protocols used to eradicate the MRSA was collected by assessment of both paper and electronic medical records and pharmacy records for the patients. Four out of the 20 patients were not offered eradication therapy secondary to longstanding documented non-adherence to therapy. Medical and pharmacy records for three out of the 20 patients were incomplete for treatment details and were excluded. Thirteen patients’ treatment histories were then analysed.

Approval for the Australian CF Data Registry was granted by the Western Sydney Local Health District Ethics Committee and all patients gave written informed consent. This subgroup analysis was also approved separately.

## 5. Conclusions

In conclusion, the successful eradication of MRSA is most effective in intermittently colonised subjects. In that situation, the eradication attempt is usually successful when early eradication is achieved; prolonged antibiotic therapy is of little use to achieve eradication.

## Figures and Tables

**Table 1 antibiotics-08-00113-t001:** Colonisation rates for methicillin-sensitive *Staphylococcus aureus* (MSSA), methicillin-resistant *Staphylococcus aureus (*MRSA) and *Pseudomonas aeruginosa (*PsA), (rounding to whole number values).

Organism	Chronic	Intermittent
MSSA	27%	17%
MRSA	5%	3%
PsA	58%	10%

**Table 2 antibiotics-08-00113-t002:** Demographics of cystic fibrosis (CF) patients with chronic or intermittent MRSA colonisation.

Age	Gender	Occupation	MRSA	Status	Other Chronic Organism
30	Male	Electrical Engineer	Chronic	Failed	PsA, coliform
25	Male	Builder	Chronic	Eradicated	PsA
45	Female	Unemployed	Chronic	Failed	nil
23	Male	Personal Trainer	Chronic	Not Offered	nil
47	Male	Jockey	Chronic	Failed	nil
41	Female	Bus Driver	Chronic	Failed	PsA
41	Male	Unemployed	Chronic	Eradicated	PsA
30	Male	Unemployed	Chronic	Eradicated	PsA
30	Female	Teacher	Chronic	Failed	nil
37	Male	Teacher	Chronic	Failed	PsA
23	Female	Student	Chronic	Failed	nil
29	Female	Waiter	Chronic	Failed	PsA
47	Female	Unemployed	Intermittent	Not Offered	PsA
38	Female	Retail	Intermittent	Eradicated	PsA
24	Male	University Student/Fireman	Intermittent	Eradicated	nil
30	Male	Administration	Intermittent	Eradicated	PsA
27	Male	Support Assistant	Intermittent	Not Offered	nil
29	Female	Unemployed/ Mother/Childcare	Intermittent	Not Offered	nil
30	Female	Psychologist	Intermittent	Eradicated	PsA
30	Female	Nurse	Intermittent	Eradicated	PsA

**Table 3 antibiotics-08-00113-t003:** Comparison of successful and failed MRSA eradication in those with complete data set.

Variable	Eradicated	Failed
Number of Adults	6	7
Average Age (years)	28.83	26.7
Number receiving vancomycin prior to eradication	5	6
Average number of courses of vancomycin	4.2	10.3
Average course length of vancomycin (weeks)	15	23.3
Average number of rifampicin / fucidin courses	1.5	1.9
Average duration of courses of rifampicin / fucidin (months)	4.3	9.9
PsA co-infection	100%	57.1%

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
