# Peer review of "Intermittent colonisation with Methicillin-Resistant Staphylococcal aureus can be eradicated from the Airways of Adults with Cystic Fibrosis"

_antibiotics, 2019, doi:10.3390/antibiotics8030113_

Round 1

Reviewer 1 Report

The manuscript by Ranzenbacher et al. described a summary on MRSA colonisation in patients with cystic fibrosis. By examining 20 patients' demographics, they conclude that eradication is more likely to be successful in intermittently colonized patients, in early attempts, while prolonged Rif/Fus antibiotic treatment does not have significant effectiveness. The manuscript is well written and nicely presented. I do not have major issues with the paper. Some minor points have been listed below,

- P1 L20, three pathogens are listed here, 'both' is not appropriate here.

- P1 L34, typo. should be 2001, not 20001 

- P2, L41, Pseudomonas aeruginosa should be italic

Author Response

Agree with all suggested edits, same actioned.

Reviewer 2 Report

Major comments: This manuscript describes MRSA eradication in an adult CF centre. Although the manuscript has been clearly written, attention needs to be given the bacterial nomenclature throughout. The introduction would benefit from further information from the literature about the advantages and challenges of eradication therapy. In addition, the discussion needs to be expanded. The authors have not discussed other publications and/or guidelines for the eradication of MRSA in this patient group in any detail, and the interplay between Pseudomonas aeruginosa and Staphylococcus aureus should also be expanded upon based on the literature.

Minor comments:

Title: Please change “Staphylococcal Aureus” to “Staphylococcus aureus

Abstract: Staphylococcus aureus, please correct.

Line 34: 2001, rather than 20001

Line 35: Chronically and intermittently colonised/infected. Please change this sentence.

Line 38: Please give the table title more detail

Line 40: it would be good to have the numbers of patients who were intermittently colonised whose organisms were eradicated in the main text.

Line 41: P. aeruginosa or PsA

Line 43: Please change “haemophilus influenza” to “Haemophilus influenzae”

Line 44: Change pseudomonas to Pseudomonas

Lines 51 and 52: Change to “Eradication was successful in…”

Table 2 title: The title does not really describe what is in the table.

The table does not make it clear which regimens were oral and which were IV.

Line 60: Please add the reference.

Author Response

Major comments: This manuscript describes MRSA eradication in an adult CF centre. Although the manuscript has been clearly written, attention needs to be given the bacterial nomenclature throughout. The introduction would benefit from further information from the literature about the advantages and challenges of eradication therapy. In addition, the discussion needs to be expanded. The authors have not discussed other publications and/or guidelines for the eradication of MRSA in this patient group in any detail, and the interplay between Pseudomonas aeruginosa and Staphylococcus aureus should also be expanded upon based on the literature.

-Some detail on eradication therapy success added to the introduction.

-Further detail regarding success rates, challenges and PsA colonisation added to discussion and thus not repeated in the introduction/

Minor comments:

Title: Please change “Staphylococcal Aureus” to “Staphylococcus aureus” -done

Abstract: Staphylococcus aureus, please correct. -done

Line 34: 2001, rather than 20001 -done

Line 35: Chronically and intermittently colonised/infected. Please change this sentence. -done

Line 38: Please give the table title more detail -done

Line 40: it would be good to have the numbers of patients who were intermittently colonised whose organisms were eradicated in the main text. -text describing patients and eradication expanded.  Extra table added to further clarify MRSA, MSSA and PsA rates.

Line 41: P. aeruginosa or PsA -done

Line 43: Please change “haemophilus influenza” to “Haemophilus influenzae” -done

Line 44: Change pseudomonas to Pseudomonas -done

Lines 51 and 52: Change to “Eradication was successful in…” -done

Table 2 title: The title does not really describe what is in the table. -titled updated and more detail added

The table does not make it clear which regimens were oral and which were IV. -Clarified in the paragraph reflecting All Vancomycin IV and All Rif/Fus PO

Line 60: Please add the reference. -updated text to include reference

Reviewer 3 Report

The manuscript “Eradication of methicillin-resistant Staphylococcus aureus from the airways of adults with cystic fibrosis” by Ranzenbacher et at. describes the statistics on the number of cystic fibrosis patients at the Westmead Adult CF Service colonized with MRSA and the success rates of MRSA eradication by standard treatment regimens depending on the colonization type (intermittent or chronic).

The presented results constitute an important piece of data and are adequately described. However, some improvements to the text could be made.

1)      It would be interesting to compare the MRSA eradication efficiency with that of MSSA. I would recommend adding the data on MSSA colonization (both the general statistics on the intermittent and chronic colonization and the eradication efficiency), if possible. Even the brief statement on the amount of MSSA colonized patients and the percent of successfully eradicated cases would be valuable.

2)      The discussion lacks a comparison with the existing literature. It would be helpful for the readers if the results were discussed in the context of the existing literature on the success rates of MRSA eradication in CF patients.

3)      The observation period is unclear. It is stated to be 2001-2015 at lines 28 and 80, but 2001-2018 at line 34.

Minor typos: staphylococcus (the first letter should be capital) at line 13, 20001 at line 34, and mucopirocin/mucopiricin at lines 25 and 74 (the antibiotic is called mupirocin).

Author Response

1)      It would be interesting to compare the MRSA eradication efficiency with that of MSSA. I would recommend adding the data on MSSA colonization (both the general statistics on the intermittent and chronic colonization and the eradication efficiency), if possible. Even the brief statement on the amount of MSSA colonized patients and the percent of successfully eradicated cases would be valuable.

-Im afraid that this may form another retrospective study altogether.  I have included the MSSA and PsA colonisation data in the form of Table 1.  I do not have MSSA clearance rates easily available for comment within the 10day manuscript re-submission period (I would need to access medical records both electronic and paper for 110 patients), however do believe this would form valuable literature in the future.

2)      The discussion lacks a comparison with the existing literature. It would be helpful for the readers if the results were discussed in the context of the existing literature on the success rates of MRSA eradication in CF patients.

I have now included references to clearance rates in 3 studies previously published.

3)      The observation period is unclear. It is stated to be 2001-2015 at lines 28 and 80, but 2001-2018 at line 34.

2018 typo corrected

Minor typos: staphylococcus (the first letter should be capital) at line 13, 20001 at line 34, and mucopirocin/mucopiricin at lines 25 and 74 (the antibiotic is called mupirocin).

These all corrected

Round 2

Reviewer 2 Report

The authors have addressed all my queries and suggested changes.